# The Effect of a 14-Day *gymnema sylvestre* Intervention to Reduce Sugar Cravings in Adults

**DOI:** 10.3390/nu14245287

**Published:** 2022-12-12

**Authors:** Sophie Turner, Charles Diako, Rozanne Kruger, Marie Wong, Warrick Wood, Kay Rutherfurd-Markwick, Eric Stice, Ajmol Ali

**Affiliations:** 1School of Sport, Nutrition and Exercise, Massey University, Auckland 0632, New Zealand; 2School of Food and Advanced Technology, Massey University, Auckland 0632, New Zealand; 3School of Health Sciences, Massey University, Auckland 0632, New Zealand; 4Centre for Metabolic Health Research, Massey University, Auckland 0632, New Zealand; 5Department of Psychiatry and Behavioral Sciences, Stanford University, Stanford, CA 94305-5717, USA

**Keywords:** sugar reduction, sensory evaluation, sweet food-food frequency questionnaire, health, behaviour change, hedonic

## Abstract

Gymnemic-acids (GA) block lingual sweet taste receptors, thereby reducing pleasantness and intake of sweet food. Objective: To examine whether a 14-day gymnema-based intervention can reduce sweet foods and discretionary sugar intake in free-living adults. Healthy adults (n = 58) were randomly allocated to either the intervention group (INT) or control group (CON). The intervention comprised of consuming 4 mg of *Gymnema sylvestre* containing 75% gymnema acids, a fibre and vitamin supplement, and an associated healthy-eating guide for 14 days; participants in the CON group followed the same protocol, replacing the GA with a placebo mint. Amount of chocolate bars eaten and sensory testing were conducted before and after the 14-day intervention (post-GA or placebo dosing on days zero and 15, respectively). Food frequency questionnaires were conducted on days zero, 15 and after a 28-day maintenance period to examine any changes in intake of sweet foods. A range of statistical procedures were used to analyse the data including Chi square, *t*-test and two-way analysis of variance. Post dosing, INT consumed fewer chocolates (2.65 ± 0.21 bars) at day zero than CON (3.15 ± 0.24 bars; *p* = 0.02); there were no differences between groups at day 15 (INT = 2.77 ± 0.22 bars; CON = 2.78 ± 0.22 bars; *p* = 0.81). At both visits, a small substantive effect (r < 0.3) was observed in the change in pleasantness and desire ratings, with INT showing a slight increase while CON showed a small decrease over the 14-day period. No differences were found in the intake of 9 food categories between groups at any timepoint. There were no differences in consumption of low sugar healthy foods between visits, or by group. The 14-day behavioural intervention reduced pleasantness and intake of chocolate in a laboratory setting. There was no habituation to the mint over the 14-day period. This study is the first to investigate the effect of longer-term gymnema acid consumption on sweet food consumption outside of a laboratory setting; further research is needed to assess how long the effect of the 14-day intervention persists.

## 1. Introduction

Sugar intake is increasing globally due to changing dietary patterns, such as increased availability of sweet, highly processed foods [1]. Excessive sugar intake has been linked to dental caries [2], metabolic syndrome [3], and cardiovascular disease [4], with emerging links to increased cancer risk, particularly for breast cancer [5]. Globally, most research in this area is focused on product reformulation, sugar taxation and food labelling [6,7,8,9], with few investigations evaluating dietary strategies for reducing excess sugar consumption at an individual level.

Gymnemic acids isolated from the *Gymnema sylvestre* (GS) plant native to India has been reported to exhibit anti-diabetic properties and to normalize blood sugar levels by decreasing plasma glucose and increasing insulin secretion [10,11]. Gymnemic acids selectively and temporarily suppress taste responses to sweet compounds without affecting the perception of other taste elements (salty, sour, bitter and umami) [12]. Gymnemic acids bind to taste type 1 receptors (T1R) 2 and 3 on the tongue and palate, preventing binding of sugar molecules and preventing subsequent firing of the chorda tympani nerve which sends sweet taste signals to various regions of the brain [13,14,15]. The sweet taste suppression effect of gymnemic acids is transient, generally lasting 30 to 60 min [16].

Recent experiments using formulated gymnema-containing products such as a dissolving tablet or “lozenge” found that consumption of gymnema in this form acutely reduces both the intake and pleasantness of confectionary [17,18]. Furthermore, Turner et al. [19] found that intake of a GS-containing mint significantly reduced intake of high-sugar sweet foods compared to a placebo and resulted in a decrease in the pleasantness and desirability rating of eating high-sugar sweet foods. Another key finding was that having a self-reported sweet tooth (relative to non-sweet tooth) resulted in a significant decrease in pleasantness and desire for more high-sugar sweet food after the GS mint compared to the placebo mint [19]. A small, yet growing body of research has studied the acute effects of gymnemic acids on pleasantness, desire, and sweet food consumption, however research is lacking on the effects of the longer-term impact of gymnema-containing products on intake and enjoyment of sweet foods. To our knowledge, previous studies have only investigated longer term gymnema consumption on blood glucose levels [20,21]. Similarly, very little is known about the effect of gymnema-containing products on sweet food choices outside of a laboratory environment.

The incentive sensitization model posits that habitual intake of high-sugar foods results in an elevated response of brain reward regions to cues that are repeatedly associated with intake of such foods, and that this elevated reward region response to food cues drives overeating [22]. Consistent with this theory, elevated reward region response to high-calorie food images and cues predicts future weight gain [23,24,25,26]. Critically, a randomized trial found that an acute dose of GS reduced fMRI-assessed brain reward region response to a cue that signaled impending tastes of a high-sugar beverage [27]. Moreover, GS mint intake reduced reward region responses to the taste of high-sugar beverages, providing evidence that GS administration reduces reward region response to food cues. An intervention that reduces habitual intake of high-sugar foods should theoretically reduce elevated reward region response to cues for high-sugar foods and reduce future dietary sugar intake.

The primary objective of this study was to assess the effectiveness of a 14-day GS supplementation package in reducing perceived ratings of desire for sweet foods, pleasantness ratings of sweet foods; the secondary objective was to assess actual consumption of sweet foods over a 14-day period.

## 2. Methods

### 2.1. Study Design

This study used a mixed methods design to examine the effects of a double-blind randomized control trial on the effectiveness of a 14-day programme to reduce intake of high sugar, discretionary foods in healthy New Zealand adults. A priori hypotheses and analyatical data plan were specified before commencement of data collection as per ethics application requirements. Ethical approval was granted by the Massey University Human Ethics Committee: Northern (Application NOR 19/52). This trial was registered with the Australian and New Zealand Clinical Trials Registry (ACTRN12619001558112) from http://www.anzctr.org.au/ (accessed on 12 November 2019). The results of the qualitative approach to explore and better understand participants’ experiences with sugar cravings, and the effects of the 14-day intervention on their relationship with sugar will be reported elsewhere (manuscript in preparation).

The 14-day programme consisted of (i) a mint (dissolving tablet) to be consumed three times per day, between meals or when craving sweet foods; (ii) a water-soluble vitamin and fibre blend to be taken early in the morning and (iii) a guide to healthy eating. For the intervention (INT) group, the mint contained 4 mg of *Gymnema sylvestre* (75% gymnemic acids), the powder contained a prebiotic fibre blend (vitamin D, thiamin, vitamin B6, vitamin B12, magnesium, zinc and chromium) and the healthy eating guide was “Eat, Treat, Delete” by Harley Pasternack [28]. The vitamin and fibre blend was included in the intervention due to concerns that consuming the mint would cause participants to reduce their intake of sweet-tasting foods that provide significant nutritional value (e.g., fruit, milk or milk products). The CON group consumed an isocaloric placebo mint and powder and used a healthy eating guide that was a readily available resource for New Zealand adults published by the Ministry of Health (“Healthy Eating, Active Living”) [29]. Participants completed a daily diary to monitor their adherence to the programme (indicating if and when they consumed each mint or sachet), with room for additional comments if they noticed any effects as a result of the programme (e.g., changes to eating habits, cravings, bowel movements, etc.).

### 2.2. Participants

Focusing on a mean difference of 0.5 in hedonic scores (sensory test) to be indicative of significant difference (as the midpoint on the scale between anchors was rounded up to the nearest whole number for interpretative purposes), the sample size estimate was 52 using 5% significance level, 90% power, and effect size (r) of 0.20. To account for participant drop out, 61 participants were recruited. Recruitment methods included email distribution lists to previous study participants, promoting the study on social media, word of mouth and flyers distributed around Massey University in Auckland, New Zealand. Potential participants were invited to complete an online screening questionnaire to capture any exclusionary medical conditions and their preferred chocolate selection from a range of 15 popular choices (Qualtrics, Provo, Utah). An information sheet outlining study protocols and time commitments was provided should they choose to participate. The inclusion criteria were healthy people aged 18–45 years living in Auckland. Exclusion critera were diagnosed diabetes, coeliac disease, gluten intolerance, or having a pacemaker. Those who satisfied the inclusion/exclusion criteria were emailed to find a suitable appointment time and to further explain their commitment to take part in the study over a six-week period. Initially, 129 potential participants completed the screening questionnaire, of which five were excluded for not meeting the inclusion criteria, one declined to participate and 63 were unable to attend the laboratory sessions. After the first visit, two participants (both from the CON group) dropped out due to unrelated medical issues; one participant (CON group) was lost to follow up after the second visit (Figure 1 outlines participant recruitment and follow up). Subsequently, 58 healthy adults (33% men and 67% women) completed this study (mean age = 29.5 years, BMI = 26.1 ± 6.6; Table 1). There were no differences between participants in the intervention and control groups for any of the variables.

### 2.3. Research Design

Participants visited the Human Nutrition Research Unit and sensory research facilities at Massey University on three separate occasions (each approximately one hour in duration) between November 2019 and February 2020.

The first and second visits were 14 days apart (days zero and 15), and the third (final) visit was 28 days after the second visit (day 44). On day zero, participants were again presented with an information sheet and they provided written informed consent prior to commencing study activities.

At the first visit all the participants (n = 61) were randomly allocated to one of the two treatment groups based on a random number generator. The investigation team was blinded to which group participants were allocated to during the data collection and data analysis phases of the research. A researcher not involved in this study prepacked participants’ take-home packages consisting of mints, powder, healthy eating guide and adherence log book, and labeled them by treatment group.

Post-GA or placebo dosing (day zero), i.e., between day 1 (the day following the initial laboratory visit) and day 15, participants consumed the products within the appropriate pack outlined above. Participants were also instructed to return any unused mints and supplements at their day 15 visit. Fifty-eight participants returned their packaging and had consumed greater than 90% of the supplied mints and powder supplements.

### 2.4. Anthropometric Measures

Body composition data was collected using bioelectrical impedance analysis (BIA; InBody230 BIA, Biospace Co., Ltd., Seoul, Republic of Korea). A stadiometer (Seka 213, Sweden) was used to measure participants’ (double measured) height and entered into the BIA machine. Participants were instructed not to drink any fluids for the one hour leading up to the scheduled visit and to empty their bladder immediately prior to undergoing BIA measurement.

### 2.5. Sensory Testing

Sensory evaluation was carried out on days zero and 15 in individual booths under red light to mask the colour of the mints. Forty-five minutes prior to sensory testing, participants consumed one standard serving (22 g, 438 kJ) of wholegrain chips (Grainwaves^®^, Bluebird Foods, Auckland, New Zealand) to reduce and standardize hunger [17].

Participants rated their hunger (on a 100-point VAS) prior to consuming all of a standardized serving (i.e., manufacturer packaging) of their favourite chocolate (chocolates varied between 14–18 g; energy varied between 292–370 kJ; Table 2). Scale data were collected electronically using Compusense Cloud (Guelph, ON, Canada) sensory software via tablets (iPad, Apple Inc., Cupertino, CA, USA).

Following the chocolate serving, participants rated the pleasantness and their desire for further chocolate servings. They received the mint and were asked to place it on their tongue, moving it around their mouth until it dissolved completely. Participants in the INT group received the gymnema-containing mints (GS mint; “Sweetkick”), and those in the CON group were given the isocaloric placebo; both provided by Nu Brands Inc. (Los Angeles, CA, USA). Participants waited 90 s after the complete dissolution of the mint and then again rated their hunger level, ate a second chocolate serving, then rated the pleasantness of the chocolate and desire for more servings again.

Hunger, pleasantness and desire for another serving were each assessed using the 100-point VAS. Hunger was rated from “I am not hungry at all” to “I am extremely hungry”, 30 s prior to each serving. Pleasantness, anchored by “Not at all pleasant” and “Very much pleasant”, was rated immediately following chocolate consumption, as was the desire for another chocolate serving (“No, not at all” to “Yes, very much”). The first two chocolate servings (pre-mint and immediately post mint) were compulsory, but for all subsquent chocolate participants indicated whether they would like another serving of chocolate (yes/no answer). If participants chose “no”, sensory testing ceased. If participants selected “yes”, the above procedure was repeated for a maximum of six chocolate servings. This followed the detailed sensory testing methods described in Turner et al. [19].

### 2.6. Questionnaires

The sweet-food food frequency questionnaire (SF-FFQ) was developed to capture the frequency of sweet food and sweetened beverage intake over the past month [30]. The SF-FFQ determines the frequency of participants’ habitual intake of 69 specific sweet-tasting foods, including both natural sweet-tasting foods and processed sweetened foods within the following categories: fruits and vegetables (n = 20); dairy-based products (n = 4); cereals (n = 5); cakes, biscuits, sweet foods (n = 14); desserts (n = 6); spreads and sweeteners (n = 6); and beverages (n = 15). All items were scored for frequency of use: never, less than once a month, 2–3 times per month, once per week, 2–4 times per week, 4–6 times per week, once a day, and twice or more a day.

Eleven further open-ended questions were asked to elicit participants’ favourite foods (to establish an understanding of whether they prefer sweet or savoury foods), number of teaspoons of sugar they added to cereal and hot beverages, and whether they snack between meals. Participants also self-reported whether they considered themselves to have a “sweet tooth”.

### 2.7. Data Handling

BMI was calculated using the Quetelet index (weight (kg)/height (m)^2^) and reported as a continous variable. The Sf-FFQ data was downloaded from Qualtrics as a spreadsheet (Excel, Microsoft Office 365, Redmond, WA, USA) and checked for completion. The consumption frequencies of each of the 69 food items were converted to a daily frequency equivalents (DFE) by allocating proportional values to the frequency of consumption options calculated with reference to a base value of 1.0 for foods eaten “once a day” [31]. DFE scores were reported as mean ± SD and a mean DFE was calculated for each food category [31,32]. All food items were categorised as either everyday foods (to be consumed regularly every day) or occasional foods (to be consumed irregularly or occasionally). The “everyday foods” category (n = 20), consisted of foods which should be eaten daily as part of a balanced diet including fresh fruit, vegetables, unsweetened milk and dairy products, plain cereals based on the Eating and Activity Guidelines for New Zealand Adults [33]. Occasional foods (n = 49) included foods that are high in fat, sodium and sugar, and/or poor sources of micronutrients and/or highly processed such as dried fruit, sweetened dairy products, high sugar cereals and spreads which should not be consumed on a regular basis as part of a balanced diet.

### 2.8. Statistical Analysis

Chi-square analysis was used to determine significant differences in the number of participants that declined further servings of the confectionery at each serving point. Average number of servings received was compared between the CON and INT groups using the Mann–Whitney U test within visits and the Wilcoxon signed-rank test used to compare this difference between groups. Data from participants who declined servings of chocolate after consumption of the mint were treated as missing data. The resulting unbalanced repeated measures data were analyzed using linear mixed effect model using Restricted Maximum Likelihood (REML) with Kenward-Roger degrees of freedom correction. The least square means and standard errors for hunger, pleasantness, and desire ratings for the INT and CON groups at each of the serving points (servings 1–6) were summarized to show difference in ratings at each serving point. Standard error was reported to show the spread of the reported scores relative to the true population mean. Since the data for serving 2 were collected just after consuming the mint, these results could be regarded as the immediate effect of the mint consumption. Subsequent analysis focused on servings 1 and 2 comparing changes in ratings before and after consumption of the mint within visits and between visits. A two-way ANOVA was conducted for demographic variables, sweet tooth status and mint type to determine their effects on the ratings for hunger, pleasantness and desire using the SAS proc mixed procedure. Significant differences were established at *p* < 0.05. Correlation effect sizes (r) were calculated using the t-statistic and degrees of freedom from each test. For the repeated measures ANOVA analysis, effect sizes were calculated using the compute.es package in RStudio (Del Re, 2013). An effect size (r) of 0.1 indicated a small effect, a value of 0.3 indicated a medium effect and a value ≥ 0.5 indicated a large effect [34]. Data analysis was undertaken using SAS (version 9.4, Cary, NC, USA), R (RStudio, Boston, MA, USA) and XLSTAT (version 2021.3.1, Addinsoft, Paris, France).

## 3. Results

### 3.1. Consumption, Pleasantness and Desire for Eating Sweet Foods

Figure 2A,B show the percentage of participants receiving servings of chocolate at each of the six chocolate serving times.

All participants rated their hunger levels, pleasantness and desire for the next chocolate serving before serving 1 (before mint) and serving 2 (after mint). After taking the mint, the number of participants selecting additional servings declined significantly (*p* < 0.0001), as subsequent servings were optional (servings 3–6).

After GA and placebo dosing on day zero, fewer chocolate bars were consumed by participants in the INT group (2.65 ± 0.21 bars) than the CON group (3.15 ± 0.24 bars; *p* = 0.022). However, the same difference was not seen on day 15; after the intervention, there was no significant difference in the additional number of chocolate bars eaten between the INT (0.77 ± 0.22 bars) and CON (0.78 ± 0.22 bars; *p* = 0.800) groups. The CON group consumed significantly more chocolate at day zero (1.15 ± 0.24 bars) than day 15 (0.78 ± 0.22 bars; *p* = 0.018), whereas there was no significant difference in the amount of chocolate eaten compared with the INT group at these two time points (*p* = 0.49); specifically, 0.65 ± 0.21 bars on day zero and 0.77 ± 0.22 bars on day 15.

The least square means and standard error from repeated measures ANOVA for hunger, pleasantness, and desire for next serving obtained for CON and INT groups at each serving point for day zero and day 15 are presented in Figure 3 and Figure 4, respectively. No significant difference was observed between treatment groups and servings for hunger ratings (Figure 3A). However, significant differences were observed for pleasantness ratings (*p* < 0.0001; Figure 3B) and desire for next serving ratings (*p* = 0.0471; Figure 3C). Serving two consistently showed the largest difference in ratings between the treatment groups for pleasantness and desire for next-serving ratings. As the data were collected just after consuming the mint for serving two, these results could be regarded as the immediate effect of the mint consumption; the INT group had significantly lower pleasantness and hunger ratings compared to the CON group (*p* < 0.05).

After the 14-day intervention (day 15), hunger ratings across servings remained non-significant (*p* = 0.7211) between the treatment groups (Figure 4A) and the levels were comparable to those observed on day zero. Pleasantness ratings were significantly different between INT and CON (*p* ≤ 0.001). Following the trend observed on day zero, a difference in the pleasantness rating was observed at serving two with INT reporting a lower rating than CON (Figure 4B). However, there was no difference (*p* = 0.60) between the treatment groups for desire rating at all the serving points; although INT reported a relatively lower rating at serving two compared to CON (Figure 4C). Since most of the differences observed were between servings one and two, subsequent analysis was focused on these two serving points. Changes in ratings were obtained by subtracting ratings for serving one from serving two, to reflect the net negative difference (if present).

Figure 5 shows the decrease in pleasantness and desire ratings after the participants consumed the mint from baseline to end. The INT group had a bigger change (decrease) in pleasantness ratings than the CON group (Figure 5A). However, there was little change between day zero and day 15 for both groups which was reflected in a small effect size (r = 0.11; *p* = 0.40). The results for desire ratings followed a similar trend as was observed for pleasantness ratings. The CON group had a slight decrease in desire for next serving from Day 0 to day 15m, while the intervention group showed a slight increase within the same time period. The effect size for desire for next rating was twice the effect reported for the pleasantness ratings (r = 0.22; *p* = 0.10).

### 3.2. Effect of Sweet Tooth Status on Consumption, Desire and Pleasantness Ratings

Overall, participants who reported having a “sweet tooth” showed a greater decrease in pleasantness ratings (*p* = 0.047, r = 0.27) and desire for further servings of chocolate (*p* = 0.049, r = 0.26) after consuming the mint (Figure 6A,B) than those who did not self identifiy as having a “sweet tooth”. Following the 14-day intervention, no significant differences were observed between the “sweet tooth” and “non-sweet tooth” groups in pleasantness and desire for more chocolate immediately following consumption of the mint (Figure 6C,D).

### 3.3. Effect of Gymnema sylvestre Consumption on Ad Libitum Sweet Food Consumption (SF-FFQ)

The calculated daily frequency equivalents (DFE) or daily serves of nine food categories at day 0, 15 and 44 are presented in Table 3. No significant differences in daily intake of the same nine food categories were observed between those in the INT and CON groups during the intervention period (day zero to day 15) or the maintenance period (day 15 to 44; Table 3; *p* > 0.05).

#### 3.3.1. Occasional Foods

There were no significant differences in the daily intake of 49 “occasional foods” (i.e., high-sugar snack foods like sweetened dairy products, high sugar cereals, cakes, desserts and beverages) between the INT and CON groups during the intervention period (*U* = 338, *p* = 0.210) or the maintenance period (*U* = 289, *p* = 0.069; Figure 7). However, significant within-group differences occurred over time. Within the INT group, the occasional foods intake decreased by 1.6 DFE (*p* = 0.022) and intake of cakes decreased by 0.61 DFE (*p* < 0.001); in the CON group. There was no change in intake of occasional foods (*p* = 0.694) but intake of cakes decreased by 0.08 DFE, (*p* = 0.016; (Table 3).

#### 3.3.2. Everyday Foods

There were no between-subject or within-subject differences in the median consumption of 20 everyday foods (i.e., healthy foods like fresh fruit, vegetables, unsweetened dairy products and plain cereals) between treatment groups at either the intervention period (*U* = 381, *p* = 0.559), or the maintence phase (*U* = 363, *p* = 0.522; Figure 8).

#### 3.3.3. Fruit

There were no differences in fruit intake between INT and CON groups between the intervention period (*U* = 414.5, *p* = 0.950) and the maintenance period (*U* = 370, *p* = 0.576). Median daily fruit intake increased over the course of the intervention for both INT and CON groups (Table 3). Fruit consumption increased by 0.69 serves (116% increase) per day between days zero and 15 (*p* < 0.0001) for INT, which was sustained through the maintenance period (change between days zero to 15, and days 15 to 44 was not significant, *p* = 0.086). Within the CON group, there was an increase of 0.71 DFE (153% increase) between days 0 to 15 (*p* = 0.001); but a significant decrease (*p* = 0.049) in fruit intake over the maintence period (days zero to 15 delta (Δ) change 0.030 [−0.490, 0.530]) compared to the intervention period (Δ0.530 [0.060, 1.140]).

## 4. Discussion

The aim of this study was to examine the effect of a 14-day behaviour intervention on desire and pleasantness of sweet-tasting foods (chocolate), and whether there was a reduction in sugar-sweetened food intake. The main findings are: (1) consumption of the GS-containing mint reduced the amount of chocolate bars eaten (day zero only) and pleasantness of sweet foods; (2) those who identified as having a “sweet tooth” showed a greater decrease in pleasantness ratings and reduced desire for further servings of chocolate after consuming the GS-mint; (3) there was no habituation to the mint after 14 days’ intake.

The current study supports previous research that consumption of a GS mint reduces ad libitum acute consumption of high sugar sweet foods compared to a placebo, and reduces the pleasantness of chocolate eaten and subsequent desire for further servings within a laboratory setting [18,19,27]. Interestingly, this effect was also seen on day 15, indicating that there was no habituation effect of the GS mint. These novel findings suggest that the GS-containing mint is still effective at reducing consumption after daily usage for 14 days, when participants are familiar with its sweet taste-altering effects. Although the GS mint reduced ratings of pleasantness and desire for more chocolate, there was no effect on hunger, indicating that it is the hedonic properties of the sweet foods that are being affected rather than the need for energy intake/satiety. Moreover, those who self-identified as having a “sweet-tooth” showed a greater reduction in pleasantness and desire ratings after taking the GS mint than those who did not identify as having a “sweet tooth”, consistent with our previous findings [19].

This study presents novel findings that participants in the CON group reduced their chocolate consumption, pleasantness and desire for further serves on day 15, but not on day zero. We hypothesise that this may be an effect of knowingly taking part in a sugar reduction study. Therefore, all participants increased their awareness of sugar over this time and those in the CON group (equally wanting to reduce their sugar intake) made an association between having a mint and eating sugar, suggesting that the wider behavioural modification regime applied within both groups in the study. Furthermore, this behavioural effect was observed in both the CON and INT groups, likely due to the fact that participants were informed that they would be assigned to one of two programmes that aimed to reduce sugar consumption. Therefore, particiants may have reduced their intake of sugar-containing foods, in the absence of active taste-altering compounds, simply because they were in a sugar-reduction study, i.e., a crude measure of behaviour change. Based on their involvement in the study, it is likely that participants were in the early stages of behavioural change (contemplation/preparation [35], involving a degree of awareness regarding current dietary patterns and a desire to make change.

Over the course of the study, participants were provided with tools to reduce their sugar intake and simple guidelines to follow, resulting in small changes within each treatment group. This effect has previously been described among US and Thai college students, where participants who were contemplating reducing sugar-sweetened beverage reduction were more aware of their consumption and open to reducing sugar consumption [36]. Two focus groups each with INT or with CON participants were conducted at each time point in this study. Preliminary data from the focus groups for this study (Turner et al., unpublished) suggest that the abovementioned behavioural effect from having access to simple guidelines affecting intake occurred among participants in both groups. Participants in three of the four focus groups (post day 15) felt that they had taken part in the intervention, suggesting that the blinding was effective.

Although there were no between-condition differences for occasional food intake we report the within-condition effects to provide a comprehensive description of the change observed in the trials. A decrease in ad libitum sweet occasional food intake during the intervention period (day zero to day 15)—regardless of being in the INT or CON groups—indicated an effect from being in the study. Cakes and muffins contribute an estimated 4.7% of dietary sugar intake in the diet of New Zealanders [37]. Results from an intensive, multi-component one-year intervention aiming to reduce sugar-sweetened beverage intake in adolescents, found a reduction in intake and BMI at the end of the intervention, but no differences after a further one-year follow up [38]. The authors suggested that the intensity of the intervention may have resulted in adopting other health-promoting behaviours including decreased television viewing which may explain the lack of significance after the maintenance period [38]. In the present study, the intensive 14-day intervention may have heightened the contemplation of sweet food intake in both treatment groups, resulting in greater differences in sweet food intake than if they were not participating in the present study.

These findings confirm a previous report that the consumption of the GS mint within a behaviour modification programme was most effective on those who identified as having a sweet tooth [19]. Future efforts should focus on those with a sweet food preference as this appears to be the group that experiences the greatest benefit from consuming gymnema-containing products. Screening tools such as the Sweet Taste Questionnaire (a 12-item questionnaire to evaluate attitudes, effects and control of eating sweet foods [39], or a Sweet Taste Test assessing response to sucrose would identify this subgroup for future research [40].

Short-term, restrictive weight loss diets with strict food consumption rules, e.g., the palaeolithic diet (restriction of grains, legumes, dairy, salt and refined oil) [41], or intermittent fasting (hours of eating are restricted or food intake is reduced on specific days of the week) [42], are popular [43,44]. These diets can be effective at weight loss and dietary control in the short term, but long-term data suggests that the level of restriction is difficult to sustain beyond three to six months [43]. Therefore, a long-term restrictive diet to reduce sugar intake is unlikely to work, however, it may be more effective if accompanied by lifestyle strategies. The GS mint used in the current study (“Sweetkick”) is analogous to the use of nicotine gum or patches to help smokers wean off cigarettes; however, to our knowledge, the Sweetkick mint does not have addictive properties, unlike nicotine. Smoking cessation is commonly associated with weight gain, but the Nurses’ Health Study found smoking cessation accompanied by lifestyle modifications such as daily moderate-vigorous exercise and dietary modification (≤2 servings of unprocessed red meat; ≥5 servings of fruit and vegetables; minimal high sugar treats) resulted in lower weight gain than those who did not exercise or modify their eating habits [45]. A similar impact may be seen among those who reduce their sugar intake; however, research is lacking in this area. Ultimately, the GS mint and 14-day programme are not intended to be used on a long-term basis, but rather as a tool to increase awareness of sugar intake and to drive changes in habitual intake away from sweet-tasting, energy-dense foods. The World Health Organisation (WHO) strongly recommends reducing free or added sugar intake to less than 10% of total energy intake [46]. A recent study examining sugar intake in over 100,000 participants (French NutriNet-Santé prospective cohort study) showed that sugar intake may represent a modifiable risk factor for cancer prevention. Moreover, repeated 24-hr dietary records found significant associations with cancer risk for added sugars, free sugars, sucrose, sugars from milk-based desserts, dairy products, and sugary drinks; therefore, any intervention that can reduce sugar intake may offer considerable health benefits [5]. The methodology used in the present study focused on the frequency of consuming selected sweet foods and did not collect data on total dietary intake and therefore, we did not assess added sugar intake.

The New Zealand Ministry of Health guidelines recommend that adults should consume at least two servings of fruit per day [47]. However, the 2020/21 NZ Health Survey shows that only an estimated 48.2% of adults are achieving this guideline [48]. At baseline, participants in the present study reported consuming only about half a serving of fruit per day. The study resulted in significantly higher fruit intake in both groups. Seasonality may have had an effect on fruit intake as fruit consumption is often higher during summer months than in winter [49,50,51]. The study took place between November and February—summer months in New Zealand—and therefore, fruit consumption in both the INT and CON groups may have been higher due to seasonal fruit availabilty. A key finding from this study was that regular consumption of the GS mint did not reduce sweet fruit intake, which contributed important nutrients as part of an everyday diet. In fact, participants may have replaced their sweet treats with sweet-tasting, healthy fruit as an alternative; however, a longitudinal study would be needed to confirm this.

It is worth noting that there was a single case report of an adverse reaction to consuming GS tea thrice daily [52]; however, none of our participants complained of any adverse effects of the dosage regiment we investigated.

### Limitations and Future Directions

Participants were given instructions on how to take the Sweetkick mint (“let the mint fully dissolve on your tongue, moving it around to coat your mouth completely”), however, we did not monitor their adherence to this instruction. If the mint is not consumed as directed (e.g., chewed or swallowed), there may be reduced inhibition of sweet taste receptors on the tongue and therefore participants would retain the ability to taste sweetness. Further research should monitor the time taken to dissolve the mint to ensure the mint is taken as directed and all T1R receptors are affected by the product. We provided different healthy eating guides and sachets to both groups; future studies should examine the individual effects of the different aspects of the behavioural intervention. Although empty packaging was collected, no tools were used to confirm that participants ingested the products as instructed. Future work should involve a biometric measure to confirm consumption, e.g., inserting riboflavin into the intervention mint [53], or using glucose monitors.

Future research should explore the effect of this intervention in people who consume high amount of sweet foods, and/or with impaired glucose tolerance that are not yet taking oral hypoglycaemic agents, as *Gymnema sylvestre* is also purported to normalise blood glucose levels [54]. Further research is needed to determine how long the effect of the 14-day intervention persists. This may also assist in determining whether it would be useful to have a “reinforcement” (or “booster”) period where participants actively re-engage with the programme after a set period of time to enhance or maintain new sweet food consumption behaviours. The effect of the intervention on different user groups needs to be explored further, including (but not limited to) individuals with obesity, pre-diabetes and/or diabetes, and athletes (who may need to reduce sweet food consumption in non-competitive periods). Moreover, as sweet food intake can impact reward regions of the brain [27,55], research on whether reward region responsivity changes after prolonged GS use warrants further investigation.

## 5. Conclusions

This study aimed to examine the effect of a 14-day ‘sugar reset’ behavioural intervention on desire, pleasantness and intake of sweet foods. Consumption of the GS-containing mint reduced desire, pleasantness and intake for further sweet food. There was no habituation to the GS-mint over the 14-day period (i.e., the mint was just as effective on day 0 as it was on day 15). There was an independent behavioural effect simply by being part of the 14-day intervention. This is the most comprehensive study in this emerging research area, and the only work so far to investigate the effect of longer-term gymnema acid consumption on sweet food consumption outside of a laboratory setting.

## Figures and Tables

**Figure 1 nutrients-14-05287-f001:**
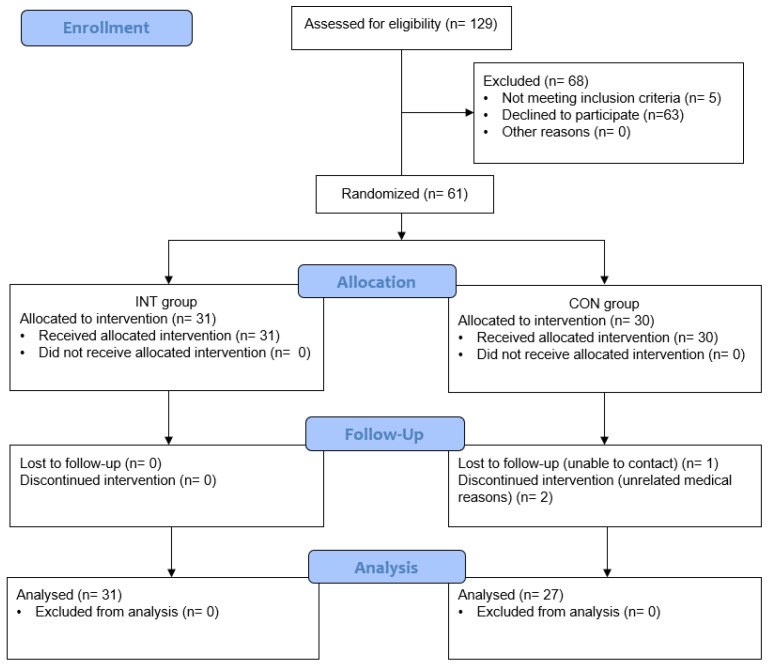
Flowchart of study recruitment and follow up.

**Figure 2 nutrients-14-05287-f002:**
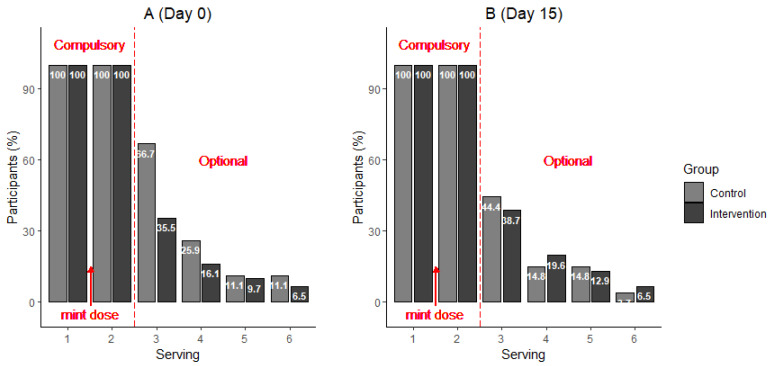
Percentage of participants receiving servings of chocolate at each of the six chocolate serving times. Participants rated their desire for each serving of chocolate after consuming the previous serving of chocolate. The first two servings (one before and one after consuming the respective mint) were compulsory; however, after the second chocolate, participants were able to indicate how much they would like another serving and then confirm whether they would like another serving. (**A**) (Day 0)—pre-intervention; (**B**) (Day 15)—post-intervention. Numbers in the bars represent percentages of participants who received the serving. Percentages are based on 31 participants in the INT group and 27 participants in the CON group.

**Figure 3 nutrients-14-05287-f003:**
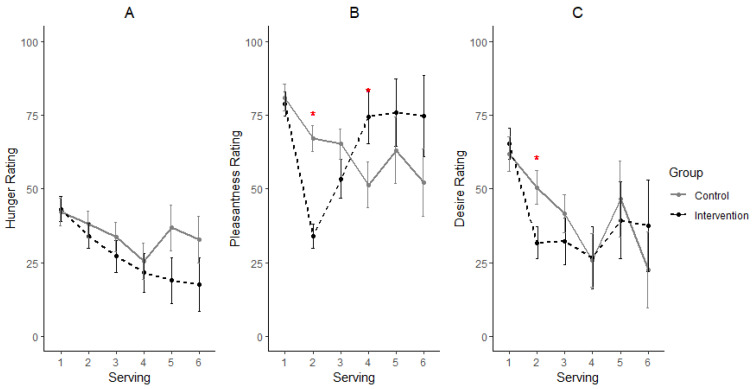
Participants’ average ratings for hunger (**A**), pleasantness of chocolate (**B**) and desire for next serving (**C**) on day 0. Means ± standard error used in the plots represent data from a total of 58, 58, 29, 12, 6 and 5 participants’ ratings at servings 1, 2, 3, 4, 5 and 6, respectively. * Significant differences at a serving (*p* < 0.05).

**Figure 4 nutrients-14-05287-f004:**
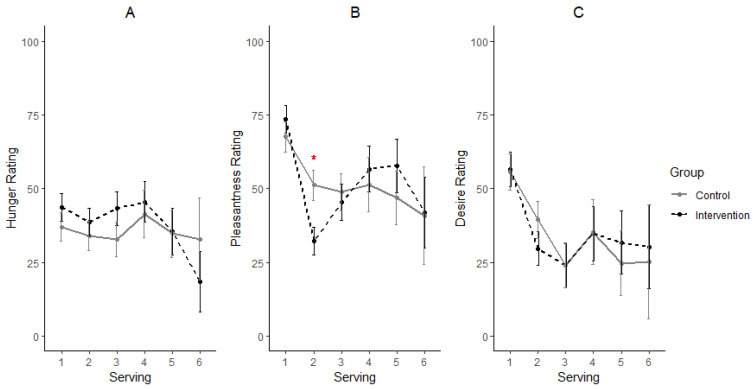
Participants’ ratings for hunger (**A**), pleasantness of chocolate (**B**) and desire for next serving (**C**) on day 15. Means ± standard error used in the plots represent data from a total of 58, 58, 24, 10, 8 and 3 participants’ ratings at servings 1, 2, 3, 4, 5 and 6, respectively. * Significant differences at a serving (*p* < 0.05).

**Figure 5 nutrients-14-05287-f005:**
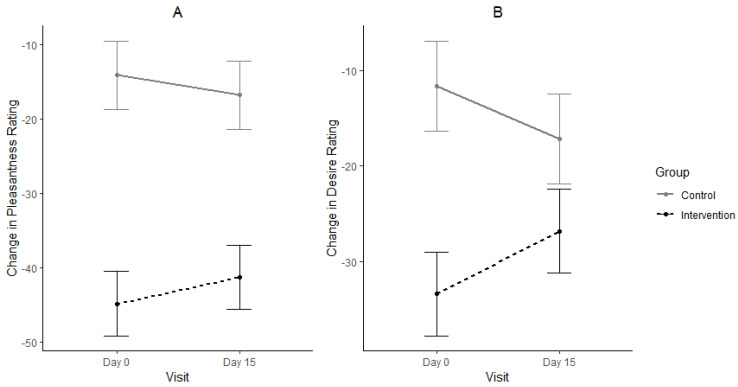
Change in ratings between serving 1 (before mint) and serving 2 (post mint) for pleasantness (**A**) and desire for next serving (**B**) for day 0 and day15 between treatment groups. Means ± standard error corresponds to the size of change in rating after taking the mint.

**Figure 6 nutrients-14-05287-f006:**
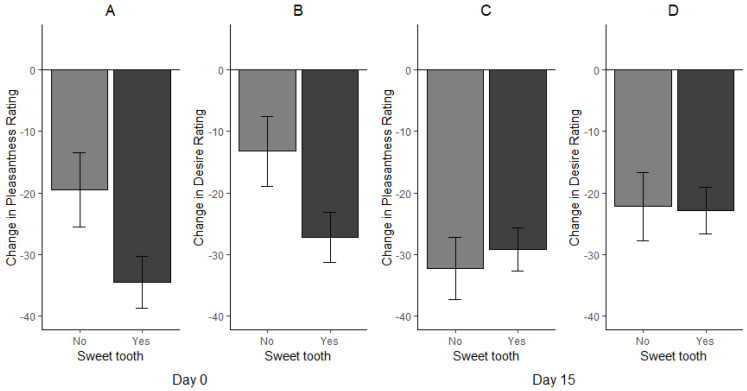
Change in pleasantness and desire ratings based on self-reported sweet tooth status for day 0 (**A**,**B**) and day 15 (**C**,**D**).

**Figure 7 nutrients-14-05287-f007:**
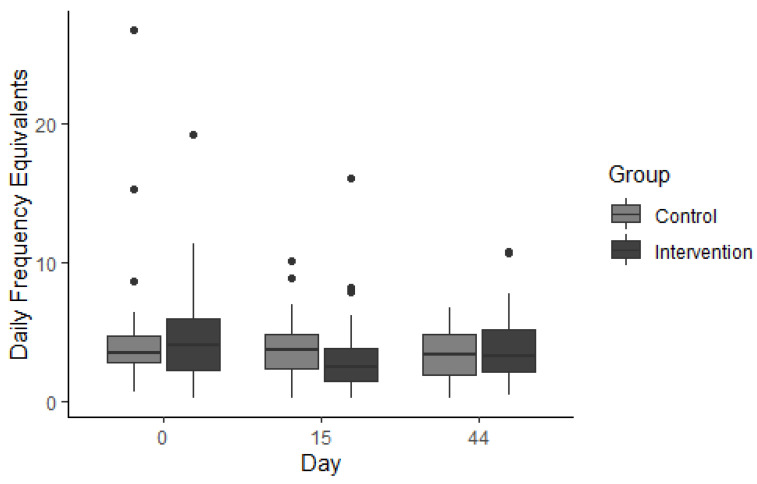
Daily frequency equivalents (DFE) of 49 “occasional” food items using the sweet foods food frequency questionnaire (Sf-FFQ). Individual dots represent outliers.

**Figure 8 nutrients-14-05287-f008:**
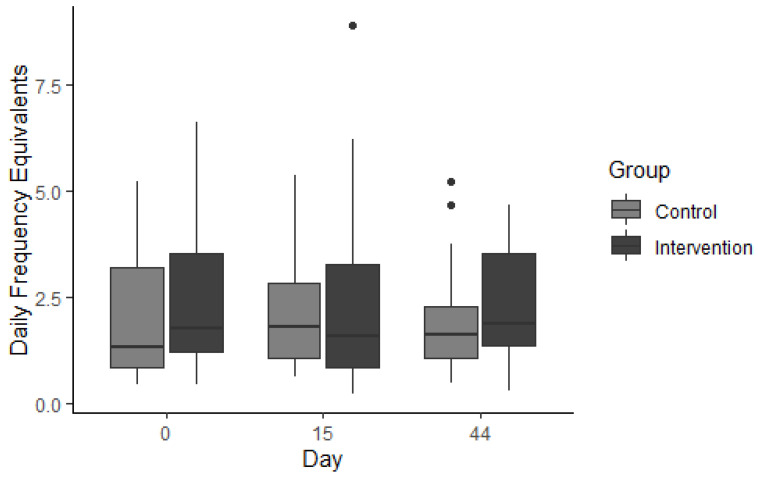
Daily frequency equivalents (DFE) of 20 “everyday” (healthy) food items using the sweet foods food frequency questionnaire (Sf-FFQ). Individual dots represent outliers.

**Table 1 nutrients-14-05287-t001:** Participant characteristics.

Characteristics (Mean ± SD) or *n* (%)	Total Group	Intervention Group	Control Group
Gender *n* (%)	58 (100)	31 (53.4)	27 (46.6)
Men	19 (32.8)	10 (52.6)	9 (47.4)
Women	39 (67.2)	21 (53.8)	18 (46.2)
Age (years)	29.5 ± 7.8	30.5 ± 7.4	28.0 ± 8.0
Median	27	33	26
Range	18–45	19–44	18–45
BMI (kg/m^2^) mean ± SD	26.1 ± 6.6	25.9 ± 7.1	26.3 ± 6.2
BMI group *n* (%)			
Underweight	2 (3.4)	2 (6.5)	0 (0)
Normal	33 (56.9)	18 (58.1)	15 (55.6)
Overweight	10 (17.2)	4 (12.9)	6 (22.2)
Obese	13 (22.4)	7 (22.6)	6 (22.2)
Weight (kg) mean ± SD	73.2 ± 19.0	72.5 ± 19.6	74.1 ± 18.6
Body fat % mean ± SD	29.2 ± 11.5	28.1 ± 10.6	30.4 ± 12.4
Ethnicity * *n* (%)			
Māori	2 (3.5)	1 (3.3)	1 (3.5)
European	34 (59.6)	20 (66.7)	14 (15.9)
Pacific Peoples	2 (3.5)	2 (6.7)	0 (0)
Asian	17 (29.8)	7 (23.3)	10 (37.0)
MELAA ^#^	6 (10.5)	3 (10.0)	3 (11.1)

* Participants were able to select multiple ethnic backgrounds and therefore the total is greater than 100%. ^#^ Middle Eastern, Latin American, and African.

**Table 2 nutrients-14-05287-t002:** Participant chocolate selection and nutrition information.

Chocolate	Participant Selection (n) * (%)		Weight (g)	Energy (kJ)	Sugar per Serve (g)	Sugar per 100 g (g)	Cocoa Content (%)
Whittaker’s^®^ Almond Gold	15	26	15	360	5.2	34.6	33
Nestle KitKat^®^	8	14	17	370	8.6	50.5	22
Whittaker’s^®^ Creamy Milk	6	10	15	352.5	6.7	44.7	33
Snickers	5	9	18	370	9.3	50.6	25
Whittaker’s^®^ Dark Peppermint	5	9	15	342	7.8	52.1	50
Cadbury Crunchie	3	5	15	292	10.3	68.7	26
Cadbury Moro Gold	3	5	17.5	327	7.3	48.6	26
Cadbury Picnic	3	5	15	327	6.8	45.6	27
Cadbury Twirl	3	5	14	316	7.7	55.2	26
Cadbury Flake	2	3	14	313	7.9	56.5	26
Twix^®^	2	3	14.5	308	7.0	48.0	25
Whittaker’s^®^ Peanut Slab	2	3	15	333	7.2	48.0	33
Mars^®^	1	2	18	350	10.5	57.1	25

* Participant section of favorite chocolate in rank order.

**Table 3 nutrients-14-05287-t003:** Daily frequency equivalents (DFE) of sweet food categories by laboratory visit (median and range).

Food Group	Intervention (DFE)			Control (DFE)		
	Day 0	Day 15	Day 44	Day 0	Day 15	Day 44
Fruit	0.59 [0.36, 1.29] ^#^	1.28 [0.68, 2.42] ^#^	1.49 [0.74, 2.35] ^#^	0.46 [0.17, 1.05] ^#^	1.170 [0.50, 2.26] ^#^	1.04 [0.54, 2.25] ^#^
Vegetable	0.30 [0.16, 0.66]	0.27 [0.14, 0.62]	0.27 [0.17, 0.66]	0.27 [0.14, 0.44]	0.28 [0.19, 0.78]	0.32 [0.17, 0.74]
Dairy	0.53 [0.08, 1.53]	0.38 [0.11, 1.03]	0.46 [0.14, 1.42]	0.50 [0.08, 1.14]	0.42 [0.08, 0.85]	0.28 [0.06, 0.85]
Cereal	0.16 [0.03, 0.59]	0.15 [0.03, 0.56]	0.16 [0.03, 0.45]	0.16 [0.03, 0.42]	0.14 [0.03, 0.66]	0.14 [0.03, 0.25]
Spreads	0.23 [0.08, 1.20]	0.17 [0.03, 0.80]	0.16 [0.03, 0.46]	0.34 [0.22, 0.70]	0.33 [0.12, 0.56]	0.22 [0.06, 0.78]
Cakes	1.31 [0.75, 2.03] ^#^	0.70 [0.42, 1.52] ^#^	1.31 [0.75, 1.03]	1.00 [0.62, 1.59] ^#^	0.92 [0.35, 1.25] ^#^	0.95 [0.41, 1.43]
Desserts	0.14 [0.08, 0.33]	0.12 [0.06, 0.22]	0.16 [0.06, 0.25]	0.16 [0.09, 0.30]	0.16 [0.08, 0.42]	0.14 [0.08, 0.44]
Drinks	0.71 [0.32, 1.15]	0.50 [0.17, 1.09]	0.47 [0.23, 1.49]	0.61 [0.33, 1.55]	0.78 [0.25, 1.91]	0.71 [0.36, 1.69]
Occasional foods ^$^	4.04 [2.22, 6.05] ^#^	2.45 [1.44, 3.87] *^#^	3.28 [2.01, 5.14]	3.45 [2.77, 4.76]	3.71 [2.37, 4.80]	3.41 [1.80, 4.76]
Everyday foods ^@^	1.79 [1.23, 3.79]	1.61 [0.84, 3.30]	1.90 [1.31, 3.58]	1.33 [0.81, 3.30]	1.83 [0.99, 2.86]	1.63 [1.06, 2.37]

^#^ Statistically significant result (*p* < 0.05) within group (Wilcoxon Signed Ranks test). * Significant result (*p* < 0.05) between groups (Mann–Whitney test). ^$^ (49 food items combined). ^@^ (20 foods combined).

## Data Availability

Data is contained within the article.

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
