# Peer review of "The Effect of a 14-Day gymnema sylvestre Intervention to Reduce Sugar Cravings in Adults"

_nutrients, 2022, doi:10.3390/nu14245287_

Round 1

Reviewer 1 Report

This manuscript presents research about the effect of Gymnema Sylvestre on acute intake of chocolate bars and how a 14-day intervention affect sweet-tasting foods. The topic is indeed interesting due to the focus on dietary strategies to reduce sugar intake, but cannot recommend the manuscript for publication as current form due to the following main reasons:

1.       In addition to the difference in GS intake, the INT and CON group also received different healthy eating guidelines. Thus, the study not only focus on the effect of GS.

2.       It is not clear, why the authors focus on chocolate in the lab study instead of focusing on the food(s) mainly contributing to sugar intake in the study population? and why they choose to focus on intake of sweet tasting food overall in the 14-day intervention without a focus on chocolate.

3.       As the authors do not use a cross-over design, careful consideration should be put in to ensuring that the two groups are similar on a number of characteristics. Most critical, the authors does not include measures to ensure that the habitual consumption of chocolate and sweet-tasting foods, respectively are the same in the two groups prior to starting the study.  

4.       In my opinion, it only makes sense to include the intensity ratings of hunger/pleasantness/desire in the cases where intake is mandatory (serving 1 and 2 on day 0 and 15, not serving 3-6). In the case of the present study, the intensity ratings are biased as to whether the participant stopped or continued their intake. Further, the sample size decreases heavily over the six servings, and reporting results of a consumer test involving as low as 5 participants does not make sense. I believe the focus on hunger/pleasantness/desire across the six servings should be removed from the manuscript.  

Also, the presentation of the studies could be more reader friendly. In general, I believe the authors should clarify that they are dealing with two types of studies: a lab study focusing on the acute effects of GS on intake and hedonic ratings of a sweet-tasting food, and a natural setting study focusing on the effects of GS on intake of sweet-tasting foods over 14-days. This division could preferably be adopted in the result and discussion section as well, including when describing the limitations.

 If the authors can provide a reply to the methodological issues which shows that a proper study design has been applied, and a willing to take on the extensive revision, I am happy to reconsider as the topic is scientifically relevant.

Author Response

This manuscript presents research about the effect of Gymnema Sylvestre on acute intake of chocolate bars and how a 14-day intervention affect sweet-tasting foods. The topic is indeed interesting due to the focus on dietary strategies to reduce sugar intake, but cannot recommend the manuscript for publication as current form due to the following main reasons:

We would like to thank the Reviewer for their critical insight of the manuscript. We have addressed all of the points they have raised, and the manuscript has been strengthened as a result.

  1. In addition to the difference in GS intake, the INT and CON group also received different healthy eating guidelines. Thus, the study not only focus on the effect of GS.

Thank you for raising this point. The intervention included the mint, the eating guide and the fibre sachet as part of a supplementation package. We have updated the wording of the objectives to reflect this issue (Lines 83-86).

  1. It is not clear, why the authors focus on chocolate in the lab study instead of focusing on the food(s) mainly contributing to sugar intake in the study population? and why they choose to focus on intake of sweet tasting food overall in the 14-day intervention without a focus on chocolate.

Previous studies have used a similar study design using chocolate bars eaten to assess volume, preference and desire for more servings (Turner at al 2020, Stice et al 2017, Nobel et al 2017). We wanted to examine the effects of the 14-day intake of the GS mint on the subsequent intake, preference and desire for more chocolate bars, and therefore we used chocolate at both time intervals (Day 0 and Day 15) to be comparable with existing research. Nevertheless, we also examined sweet food intake (“everyday” and “occasional” foods, including chocolate) during this 14-day period. As there was no statistically significant between-group differences in food intake we did not focus on this aspect in the Discussion. Moreover, we have updated the wording of the objectives to better reflect the primary focus of the study, which was the effect of the supplementation package on sensory evaluation measures (Lines 83-85).

  1. As the authors do not use a cross-over design, careful consideration should be put in to ensuring that the two groups are similar on a number of characteristics. Most critical, the authors does not include measures to ensure that the habitual consumption of chocolate and sweet-tasting foods, respectively are the same in the two groups prior to starting the study.

As recommended by methodologist we randomly assigned participants to condition because this is the best way to generate groups that are equivalent on all measured and unmeasured variables that could serve as a confound. In response, we now confirm that participants in the two groups did not differ significantly at baseline on the measured variables which we now note (Lines 139-140).

  1. In my opinion, it only makes sense to include the intensity ratings of hunger/pleasantness/desire in the cases where intake is mandatory (serving 1 and 2 on day 0 and 15, not serving 3-6). In the case of the present study, the intensity ratings are biased as to whether the participant stopped or continued their intake. Further, the sample size decreases heavily over the six servings, and reporting results of a consumer test involving as low as 5 participants does not make sense. I believe the focus on hunger/pleasantness/desire across the six servings should be removed from the manuscript.

Thank you for your comment. We did focus on serving 1 and 2 for the rest of the paper after presenting Figures 2- 4 as stated in lines 248 – 250. Therefore, we have clearly identified in Figure 2 which servings were compulsory and which were (subsequently) optional. With respect, we believe that showing how many participants are remaining after each serving is useful information relating to effect of the GS mint.

Also, the presentation of the studies could be more reader friendly. In general, I believe the authors should clarify that they are dealing with two types of studies: a lab study focusing on the acute effects of GS on intake and hedonic ratings of a sweet-tasting food, and a natural setting study focusing on the effects of GS on intake of sweet-tasting foods over 14-days. This division could preferably be adopted in the result and discussion section as well, including when describing the limitations.

We agree with the Reviewer that these aspects i.e., i) sensory evaluation on Day 0 and Day 15 and ii) sweet food intake during the 14-day supplementation period need to be more clearly outlined, and therefore the wording of the objectives has been updated to reflect this (Lines 83-85).

If the authors can provide a reply to the methodological issues which shows that a proper study design has been applied, and a willing to take on the extensive revision, I am happy to reconsider as the topic is scientifically relevant.

We thank the Reviewer for their valuable feedback and hope the points of concern have been satisfactorily addressed.

Reviewer 2 Report

This interesting trial examines the clinical anti-sweetening effect of gymnenic acids, of which the plant has longtime been used in traditional medicine. Although small, the strength of the study is the inclusion of people of multiple ethnicities. I do have the following remarks:

-          More recently, relatively important hepatic toxicity has been noticed, and the authors should comment on this.

-          The effect is relatively short-lived, meaning that intake should be done very frequently, raising patient expenses, and the authors should comment on this.

-          In the Discussion, subheading 4.1 is missing (although subheadings are usually discouraged).

Author Response

This interesting trial examines the clinical anti-sweetening effect of gymnenic acids, of which the plant has longtime been used in traditional medicine. Although small, the strength of the study is the inclusion of people of multiple ethnicities.

We would like to thank the Reviewer for their critical insight of the manuscript. We have addressed all of the points they have raised, and the manuscript has been strengthened as a result.

I do have the following remarks:

-          More recently, relatively important hepatic toxicity has been noticed, and the authors should comment on this.

We agree that it is very important to monitor for adverse reactions to any supplements like the GS-mint. In response, we now note that there was a single case report of an adverse reaction to consuming GS tea thrice daily and report that none of our participants complained of any adverse effects of the dosage regiment we investigated (Lines 499-501).

Shiyovich A, Sztarkier I, Nesher L. Toxic hepatitis induced by Gymnema sylvestre, a natural remedy for type 2 diabetes mellitus. Am J Med Sci. 2010;340:514–517.

-          The effect is relatively short-lived, meaning that intake should be done very frequently, raising patient expenses, and the authors should comment on this.

When interpreting these findings it should be noted that the GS-mints cost US$20 for 48 tablets, which would work out to be $1.26 per day (if taking 3 times daily) and $37.80 per month for a person to employ this dosage regiment.

-          In the Discussion, subheading 4.1 is missing (although subheadings are usually discouraged).

Thank you for picking this up. We have changed the numbering for the Discussion.